# Single-Atom Iron Catalyst Based on Functionalized Mesophase Pitch Exhibiting Efficient Oxygen Reduction Reaction Activity

Xianrui Gu [1], Meng Wang [2], Hongpeng Peng [2], Qian Peng [1], Wei Wang [1], Houpeng Wang [1], Junjun Shi [1], Xuetao Qin [2], Zhijian Da [1,*], Wenhong Yang [3], Yuchao Wu [1,3,*], Ding Ma [2] and Houliang Dai [1]

1 Research Institute of Petroleum Processing, Sinopec, No. 18, Beijing 100083, China
2 Beijing National Laboratory for Molecular Sciences, College of Chemistry and Molecular Engineering and College of Engineering, and BIC-ESAT, Peking University, Beijing 100871, China
3 Petrochemical Research Institute, CNPC, Block E., A42, Beijing 100871, China
* Correspondence: dazhijian@ripp.com (Z.D.); wuyuchao@petrochina.com.cn (Y.W.)

**Abstract:** Designing highly efficient and low-cost electrocatalysts is of great importance in the fields of energy conversion and storage. We report on the facile synthesis of a single atom (SA) iron catalyst via the pyrolysis of a functionalized mesophase pitch. Monomers of naphthalene and indole underwent polymerization in the presence of iron chloride, which afterwards served as the pore-forming agent and iron source for the resulting catalyst. The SA-Fe@NC catalyst has a well-defined atomic dispersion of iron atoms coordinated by N-ligands in the porous carbon matrix, exhibiting excellent oxygen reduction reaction (ORR) activity ($E_{1/2}$ = 0.89 V) that outperforms the commercial Pt/C catalyst ($E_{1/2}$ = 0.84 V). Moreover, it shows better long-term stability than the Pt/C catalyst in alkaline media. This facile strategy could be employed in versatile fossil feedstock and develop promising non-platinum group metal ORR catalysts for fuel cell technologies.

**Keywords:** functional naphthalene mesophase; single atom catalysis; ORR





## 1. Introduction

Fuel cells technologies have drawn much attention on account of their applications in the fields of energy conversion and storage [1–3]. The oxygen reduction reaction (ORR) at the cathode is an essential process, which is primarily governed by the precious-group-metal catalysts, such as platinum or its alloy [4]. However, the precious metal-based materials are limited by the high cost and low abundance in nature, impeding the commercialization of the fuel cells. Therefore, efforts have been devoted to developing non-precious metal catalysts [5,6]. Metal–nitrogen–carbon (M-N-C, M = Fe, Co) materials are promising candidates for electrochemical reactions in place of the Pt-based catalysts [7–17]. For Fe-N-C materials, nitrogen-coordinated iron atoms (Fe-N$_x$ species) dispersed in the porous carbon matrix are considered as the active sites, exhibiting high ORR activities [18–21]. The Fe-N-C catalysts are generally fabricated by pyrolysing precursors containing Fe, N and C [22,23], for example, metal complexes with N$_x$-macrocycles [24,25], nitrogen-containing polymers [3,26–28] and biomass [29]. However, these synthesis methods usually require iron salts mixing, which is time consuming and likely induces inhomogeneity to the catalysts, consequently influencing the ORR activity. Moreover, metal–organic frameworks (MOFs) [30–33] and template-assisted nanomaterials have been investigated, which could efficiently enhance the dispersity of iron atoms in the pyrolyzed matrix [20,34,35], yet these suffer from high cost and low yield; thus, they could be hardly applied to large-scale manufacturing.

To overcome the problems as mentioned above, we developed an efficient and versatile method to fabricate a single-atom iron catalyst on N-doped carbon networks (SA-Fe@NC). The precursor of the catalyst comes from a mesophase pitch, which is considered as an excellent intermediate for advanced carbon materials such as carbon foam [36], carbon fiber [37],

needle coke and C/C composites [38,39]. This pitch material is commonly obtained from the petroleum by-products with abundant aromatic fractions, for example, fluid catalytic cracking decant oil [40]. Minato et al. reported the polymerization of naphthalene into carbonaceous mesophase using a Lewis acid $AlCl_3$ as the catalyst [41]. This was further developed with more efficient catalysts and more diversed aromatic raw materials [40,42]. In this work, we synthesized the nitrogen-enriched naphthalene pitch (NP) via the co-polymerization of naphthalene and indole, which are catalyzed by the Lewis acid iron chloride. N-containing molecules could be easily oligomerized via Friedel–Crafts reaction and dispersed into the carbon matrix, introducing more functionalities to the carbonaceous precursors. Meanwhile, the iron chloride also serves as the pore-forming agent and the source of active sites in the subsequent pyrolysis process. The as-synthesized catalyst exhibits efficient ORR activity with $E_{1/2}$ (half-wave potential) of 0.89 V in 0.1 M KOH, which is 50 mV more positive than that of the commercial Pt/C ($E_{1/2}$ of 0.84 V). Moreover, it also shows better methanol tolerance and excellent stability compared to the commercial Pt/C catalyst, displaying no obvious current change in the presence of 1.0 M MeOH and little drop in $E_{1/2}$ after 5000 potential cycles. The high ORR activity is attributed to the atomically dispersed $Fe–N_x$ sites which are confirmed by X-ray photoelectron spectroscopy (XPS), high-angle annular dark-field scanning transmission electron microscope (HAADF-STEM) and extended X-ray absorption fine structure (EXAFS) results. Considering that the naturally acquired crude oil contains an abundance of heteroaromatics such as aniline, indole, pyridine and pyrrole [43], the present synthesis of SA-Fe@NC provides a novel means toward active and low-cost ORR catalysts and offers a new route for utilizing the petroleum resources.

## 2. Results and Discussion

The fabrication process of the SA-Fe@NC catalyst is shown in Figure 1. Monomers of naphthalene and indole oligomerized in the presence of $FeCl_3$, forming an NP carbonaceous precursor. This process was performed in a mild solvent-free condition and can be easily enlarged into large scale. The precursor was sampled and analyzed via Fourier transform ion cyclotron resonance mass spectrometry (FT-ICR MS). Under the atmospheric pressure photo ionization (APPI) mode, naphthalene oligomers and dehydrogenized co-polymers of indole–naphthalene can be observed between 800 and 1200 Da (Figure S1, Supplementary Material). Switching to the $ESI^-$ mode, more co-polymers of indole–naphthalene are detected as they are more likely to be ionized by the $ESI^-$ source. Patterned peaks are observed between 900 and 1200 Da on account of the length difference in co-polymers. Peaks (labeled in red) with m/z of 903.3359, 1043.3635 and 1158.4072 Da confirm the polymerization between naphthalene and indole (Figure 1). After pyrolyzing the precursor at 800 °C under $N_2$ atmosphere, followed by acid leaching, the catalyst SA-Fe@NC$_{800}$ was obtained. Brunauer–Emmet–Teller (BET) measurements were carried out and are shown in Figure 2a. The surface area and pore volume of the SA-Fe@NC$_{800}$ are 289 $m^2/g$ and 0.22 $cm^3/g$, respectively. These values are much higher than the control sample (7.29 $m^2/g$ and 0.04 $cm^3/g$) prepared with $AlCl_3$ as the polymerizing agent in replacement of $FeCl_3$ (Figure S2, Supplementary Material), suggesting the pore-forming ability of the latter during the pyrolysis process [44]. The pore size distribution curve (Figure 2a inset) also shows the abundance of mesopores in the catalyst. The graphitization degree of the catalysts was confirmed by Raman spectroscopy. In Figure 2b, two peaks are located at ca. 1374 $cm^{-1}$ and 1594 $cm^{-1}$ for the SA-Fe@NC$_{800}$, representing the D-band and G-band of graphite, respectively.

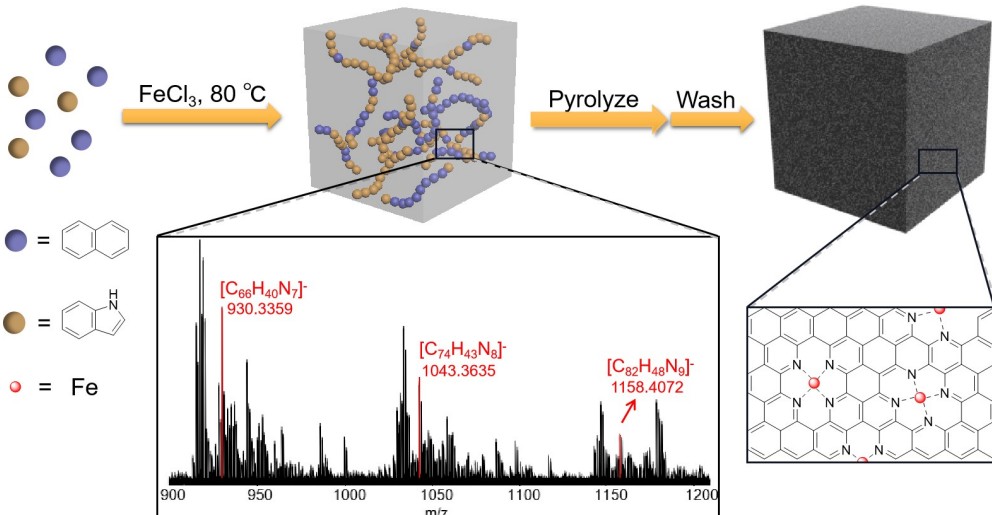

**Figure 1.** Schematic illustration of the synthesis procedure of the SA-Fe@NC.

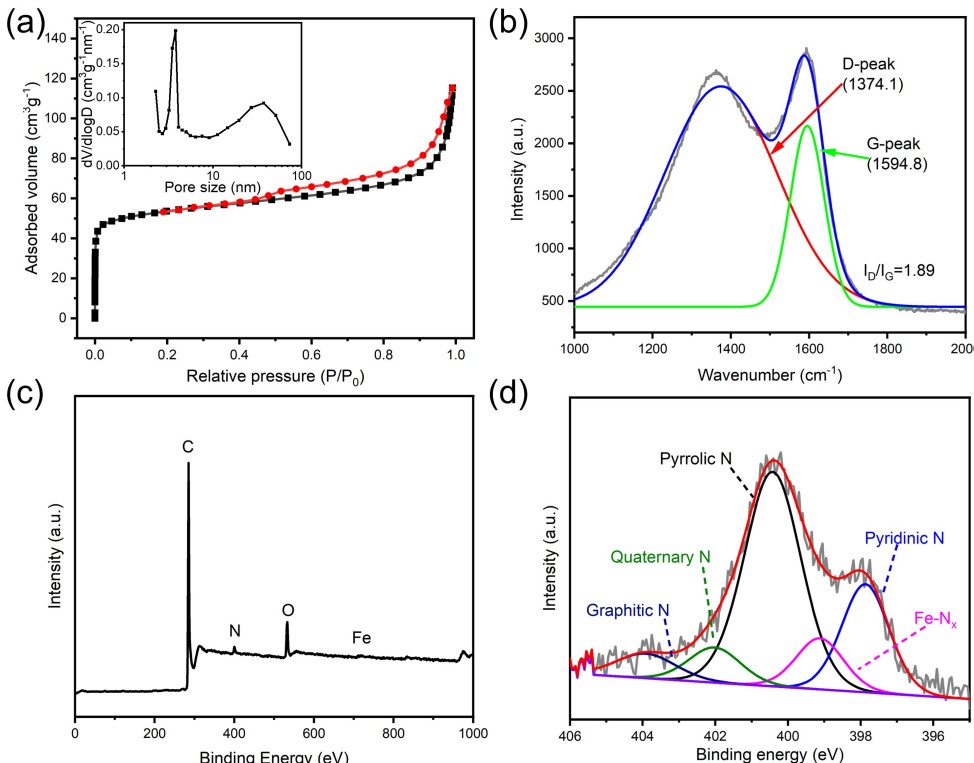

**Figure 2.** (**a**) $N_2$ adsorption–desorption isotherm (inset: pore size distribution), (**b**) Raman spectrum, (**c**) XPS survey scan spectrum and (**d**) the corresponding high-resolution N 1s spectrum of the SA-Fe@NC$_{800}$.

The corresponding $I_D/I_G$ values are employed to analyze the degree of graphitic nature of the prepared samples. As the pyrolysis temperature of the SA-Fe@NC increased from 800 to 950 °C, the $I_D/I_G$ ratio dropped from 1.89 to 1.34, indicating the formation of more ordered graphitic structures. However, when the pyrolysis temperature further increased to 1100 °C, the $I_D/I_G$ ratio augmented to 1.46, which was likely on account of the formation of defects at higher temperature (Figures S3 and S4, Supplementary Material). XPS spectrum reveals the elemental composition of the SA-Fe@NC$_{800}$ catalyst. From the survey scan of the SA-Fe@NC$_{800}$ (Figure 2c), the C, N, O and Fe elements are determined to be 90.2%, 3.72%, 5.98%, and 0.43%, respectively. The nitrogen content decreased with the

augment of pyrolysis temperature from 800 to 1100 °C (Figure S5 and Table S1, Supporting Information). When the feeding ratio between indole and naphthalene was tuned down to 1:5 during the preparation of precursor, the nitrogen content of the catalyst (SA-Fe@NC$_{800-1}$) dropped correspondingly. Yet when the indole was solely polymerized, the pyrolyzed catalyst (SA-Fe@NC$_{800-2}$) showed a significant decrease in the nitrogen content (Table S1, Supporting Information), possibly owing to the lower decomposition temperature of indole oligomers [45]. The high-resolution N 1s spectra of the SA-Fe@NCs at different temperature are further deconvoluted into five component peaks at 397.9, 399.1, 400.4, 402.1 and 403.9 eV (Figure 2d, Figures S6 and S7 in the Supplementary Material), corresponding to pyridnic N, Fe-N$_x$, pyrrolic N, graphitic N and quarternary N, respectively, and the results are summarized in Table S2, Supporting Information [46]. Among them, pyridinic N and pyrrolic N can serve as metal-coordination sites on account of their lone-pair electrons. Higher ratios of both kinds of N likely promote the gain of resulting Fe-N$_x$, which are generally considered as beneficial to the ORR [27]. The SA-Fe@NC$_{800}$ follows this trend, revealing higher proportions of pyridinic N (0.83%), pyrrolic N (1.96%) and Fe-N$_x$ (0.39%) than the rest SA-Fe@NC catalysts.

The morphology of the SA-Fe@NC$_{800}$ was investigated by scanning electron microscopy (SEM) and transmission electron microscopy (TEM). Multilayers of the catalysts are observed, stacking to form structures with hierarchical pores (Figure 3a and Figure S8 in the Supplementary Material). The mapping images reveal that C, N, and Fe elements are uniformly distributed over the entire structure, as shown in Figure 3b, which is consistent with the XPS results in Figure 2c,d. At higher magnification, Figure 3c depicts the annular bright field (ABF) STEM image. Flake-like graphitic carbon structures are clearly observed, with no existing Fe particles. The HAADF-STEM images at the same region (Figure 3d,e) demonstrate highly dispersed Fe single atoms, which are identified by the isolated bright dots marked with yellow cycles. To further confirm the existence of atomically dispersed iron atoms, the X-ray absorption near-edge structure (XANES) and EXAFS measurements were performed. The XANES spectra (Figure 4a) show that the absorption edge of the SA-Fe@NC$_{800}$ lies in between the FeO and Fe$_2$O$_3$, indicating its Fe atom valency fluctuates between Fe$^0$ and Fe$^{3+}$. In addition, the shoulder peak of the SA-Fe@NC$_{800}$ at ca. 7114 eV discloses the fingerprint of the square-planar Fe-X$_4$ moiety (D$_{4h}$ symmetry), which suggests the existence of the Fe-N$_4$ structure [47,48]. Meanwhile, the EXAFS spectra of K-edge show a peak at around 1.6 Å, and no apparent Fe-Fe coordination peak or other high shell peaks are observed (Figure 4b), indicating the formation of atomically dispersed Fe-N/O sites [18,49]. The coordination number of Fe within the Fe-N/O sites was calculated as 5 (Table S3, Supporting Information), suggesting a porphyrin planar structure of Fe-N$_4$ with an O$_2$ molecule absorbed vertically [33].

As an atomically dispersed Fe catalyst, the unique features of the SA-Fe@NCs imply themselves being ORR active. The ORR activities of the catalysts were firstly evaluated by cyclic voltammetry (CV) measurements in N$_2$- and O$_2$-saturated 0.1 M KOH solutions. In contrast to the featureless CV curves across the entire potential range in the N$_2$ solution, obvious cathodic peaks appear in the CV curves of all the catalysts in the O$_2$ solution owing to the oxygen reduction (Figure 5a and Figure S9 in the Supplementary Material). The SA-Fe@NC$_{800}$ catalyst displays a more positive cathodic peak potential (0.85 V vs. RHE (Reversible Hydrogen Electrode), the same hereafter) than the commercial Pt/C catalyst (0.82 V) and other SA-Fe@NC catalysts pyrolyzed at higher temperatures, implying a higher ORR activity. Figure 5b shows the polarization curve of the catalysts in the O$_2$-saturated 0.1 M KOH electrolyte at a rotating speed of 1600 rpm. The SA-Fe@NC$_{800}$ shows the highest half-wave potential (E$_{1/2}$) of 0.89 V and the onset potential (E$_{onset}$) of 1.06 V, which is much higher ORR activity than the SA-Fe@NC catalysts pyrolyzed at higher temperatures and the SA-Fe@NC catalysts with different compositions (Figure S10, Supplementary Material). In addition, the optimized E$_{1/2}$ and E$_{onset}$ are higher than that of the commercial Pt/C catalyst (E$_{1/2}$ = 0.84 V, E$_{onset}$ = 0.97 V).

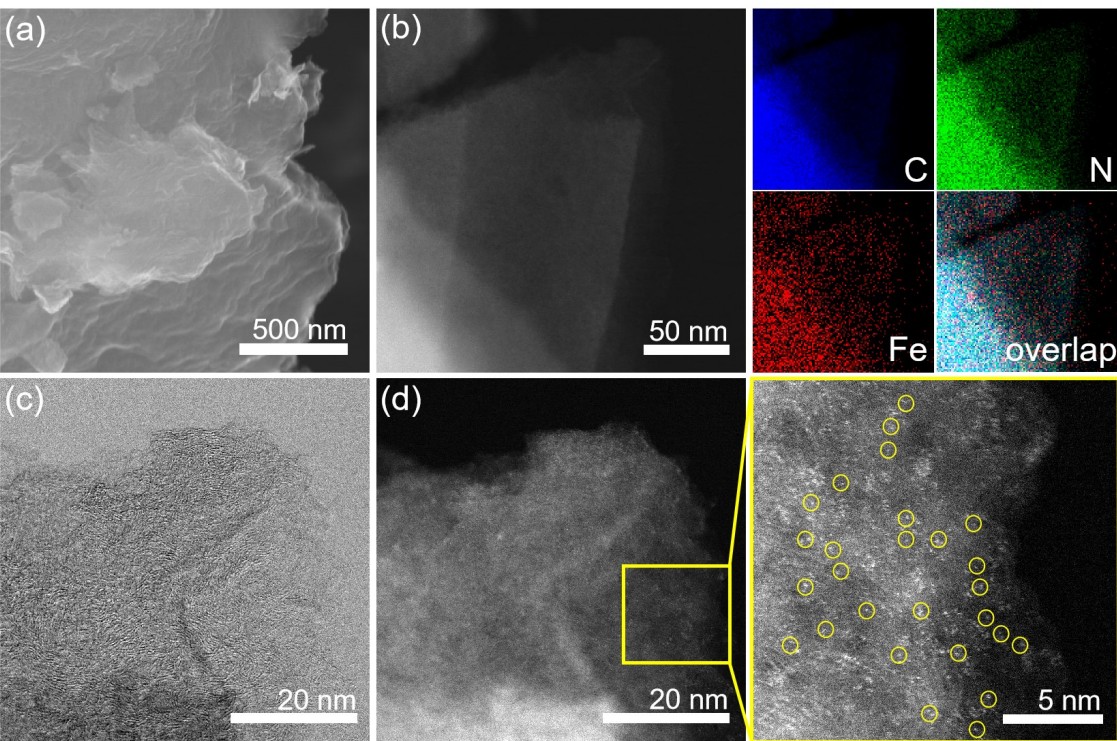

**Figure 3.** Microscopic characterization of the SA-Fe@NC$_{800}$. (**a**) SEM image, (**b**) HAADF-STEM image and elemental mapping, C (blue), N (green) Fe (red) and overlay. (**c**) ABF image, (**d**) HAADF-STEM image and the enlarged view.

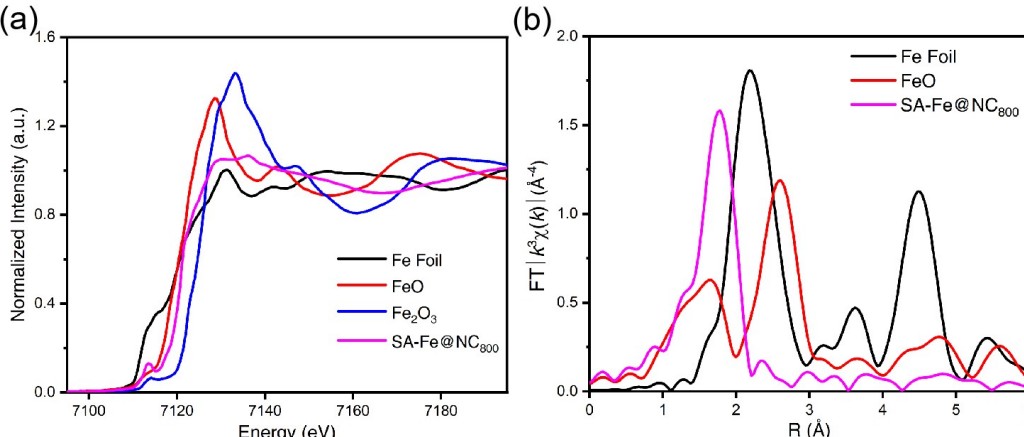

**Figure 4.** (**a**) XANES spectra at the Fe K-edge of the SA-Fe@NC$_{800}$, FeO, Fe$_2$O$_3$ and Fe foil. (**b**) Fourier transform (FT) at the Fe K-edge of the SA-Fe@NC$_{800}$, FeO and Fe foil.

As shown in Figure 5c, the RDE polarization curves of SA-Fe@NC$_{800}$ at various rating speeds are studied, and the corresponding Koutecky–Levich (K-L) plots at different applied potentials display linear relationships with the first-order reaction kinetics. The transfer electron number calculated by the K-L plots is 3.8 from a potential range of 0.5–0.7 V, indicating a dominant four-electron transfer pathway.

The ORR kinetics of the catalyst was further analyzed by Tafel plots. The tafel slope of the SA-Fe@NC$_{800}$ (86 mV dec$^{-1}$) is smaller than that of the commercial Pt/C, suggesting a faster current increasing rate in the former (Figure 5d). Furthermore, the methanol tolerance of the SA-Fe@NC$_{800}$ was evaluated by the chronoamperometric measurements in the O$_2$-saturated 0.1 M KOH electrolyte at 0.8 V with a rotation rate of 1600 rpm. A significant change in the current density is observed for the Pt/C after the addition of methanol to the electrolyte. While the catalyst SA-Fe@NC$_{800}$ shows little change in the current density

against the methanol injection across the entire observation time, indicating remarkable tolerance to methanol crossover (Figure 5e). In addition, the durability of SA-Fe@NC$_{800}$ was evaluated by cycling the catalyst between 0.6 and 1.0 V at a scan rate of 50 mV s$^{-1}$ in O$_2$-saturated 0.1 M KOH, as shown in Figure 5f. After 5000 continuous cycles, there is very little decay in E$_{1/2}$ (ca. 3 mV), demonstrating the outstanding stability of the SA-Fe@NC$_{800}$. In contrast, the commercial Pt/C is subjected to a drop of nearly 40 mV in E$_{1/2}$ (Figure S11, Supplementary Material). Meanwhile, in the long-term chronoamperometric measurements (0.8 V), the current density retains at 93.6% over 12,000 s continuous working for SA-Fe@NC$_{800}$, while the commercial Pt/C shows only 67.5% retention of the current density under the same condition (Figure S12, Supplementary Material).

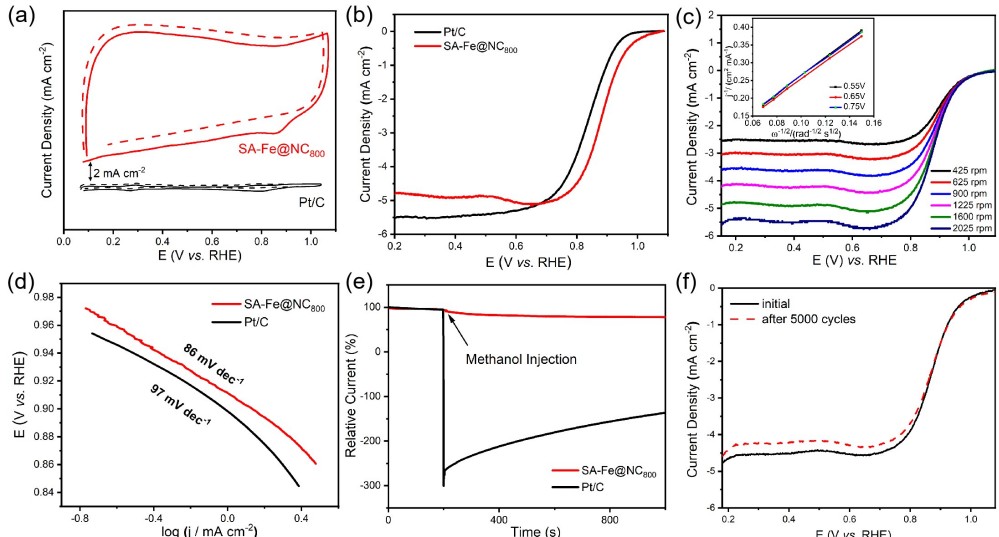

**Figure 5.** (**a**) CV curves of the SA-Fe@NC$_{800}$ and the commercial Pt/C catalysts in the O$_2$-saturated (solid line) and N$_2$-saturated (dashed line) 0.1 M KOH solutions at a sweep rate of 50 mV s$^{-1}$. (**b**) LSV curves of the catalysts in the O$_2$-saturated 0.1 M KOH electrolyte at 10 mV s$^{-1}$ with a rotation rate of 1600 rpm. (**c**) LSV curves of the SA-Fe@NC$_{800}$ in the O$_2$-saturated 0.1 M KOH solution at 50 mV s$^{-1}$ with different rotation rates (inset shows the corresponding Koutechy–Levich plots). (**d**) Tafel plots of the SA-Fe@NC$_{800}$ in 0.1 M KOH. (**e**) Methanol crossover effect test of the SA-Fe@NC$_{800}$ and Pt/C at 0.8 V. 3 mL of methanol was injected into the O$_2$-saturated 0.1 M KOH solution (90 mL) at 200 s. (**f**) LSV curves of SA-Fe@NC$_{800}$ before and after 5000 CV cycles between 0.6 and 1.0 V.

## 3. Materials and Methods

All chemicals were used as received without further purification. Hydrochloric acid (HCl, >36%), naphthalene (99%), indole (99%), and anhydrous ferric chloride (FeCl$_3$) were purchased from Sinopharm Chemical Ltd., Beijing, China, Potassium hydroxide (KOH, 99.99%) was obtained from Aladdin, Beijing, China. Commercial Pt/C (20 wt % metal) and Nafion solution (5 wt %) were supplied by Alfa Aesar. All the water in the experiments was ultrapure water (>18.2 MΩ cm).

TEM images were performed on a Technai F20 transmission electron microscope. The high-resolution TEM, HAADF-STEM images and elemental mapping were recorded on a JEOL-2100F FETEM with electron acceleration energy of 200 kV. Photoemission spectroscopy experiments (XPS) were performed at the Photoemission Endstation at the BL10B beamline. XAFS spectra at the Fe K-edge were measured at the beamline 1W1B station of the Beijing Synchrotron Radiation Facility (BSRF), China. The Fe K-edge XAFS data were recorded in a fluorescence mode. Fe foil, Fe$_2$O$_3$ were used as references. All spectra were collected in ambient conditions. The acquired EXAFS data were processed according to the standard procedures using the ATHENA module implemented in the IFEFFIT software packages (version 0.9.25, Chicago, IL, USA). The k$^3$-weighted EXAFS spectra were obtained by subtracting the post-edge background from the overall absorption

and then normalizing with respect to the edge-jump step. Subsequently, the $k^3$-weighted $x(k)$ data of Fe K-edge were Fourier transformed to real (R) space using a hanning windows (dk = 1.0 Å$^{-1}$) to separate the EXAFS contributions from different coordination shells.

### 3.1. Negative ESI FT-ICR MS Characterization

The co-polymer of naphthalene and indole was dissolved in toluene to yield a stock solution of 10 mg/mL. The stock solution was then diluted to 0.02 mg/mL in toluene/methanol (*v:v* = 1:1). Ammonia hydroxide (28%) was added to the diluted solution, and the final base concentration was 2.5% (*v:v*). The prepared sample flowed through a fused-silica capillary at a rate of 120 μL/h by a syringe pump. Analyses were conducted on a 15 T Bruker SolariX XR FT-ICR MS. The ionization source was negative mode electrospray ionization (ESI). Nitrogen (99.999%) served as the drying gas and nebulizing gas. The drying gas flow rate was 4.0 L/min at the temperature of 200 °C. The nebulizing gas flow rate was 1.0 L/min. The skimmer voltage was 15 V. The m/z range was from 50 to 3000 Da. The RF amplitudes of the collision hexapole were 800 Vpp. The time-of-flight was 1.0 ms. The data size was 8 M and time-domain data sets were 128 scans.

### 3.2. Positive APPI FT-ICR MS Characterization

The co-polymer of naphthalene and indole was dissolved in toluene to yield a stock solution of 10 mg/mL. The stock solution was then diluted to 0.02 mg/mL in toluene. The prepared sample flowed through a fused-silica capillary at a rate of 360 μL/h by a syringe pump. Analyses were conducted on a 15 T Bruker SolariX XR FT-ICR MS. Positive mode atmospheric pressure photoionization (APPI) was used as the ionization source and the light source was a Kr lamp. Nitrogen (99.999%) served as the drying gas and nebulizing gas. The temperature of the drying gas was 200 °C and the flow rate was 2.0 L/min. The nebulizing gas flow rate was 1.0 L/min, and the temperature of vaporizer was 400 °C. The skimmer voltage was 15 V. The m/z range was from 50 to 3000 Da. The RF amplitudes of the collision hexapole were 1000 Vpp. The time-of-flight was 1.8 ms. The data size was 8 M and time-domain data sets were 128 scans.

### 3.3. Electrochemical Measurement

All catalysts were prepared by mixing 6 mg of the catalyst in 1 mL of solution containing 300 μL of ethanol, 660 μL of $H_2O$ and 40 μL of 5% Nafion solution, which was followed by ultrasonication for 1 h to form homogeneous catalyst inks. Commercial 20 wt % Pt/C was prepared as a 5 mg/mL solution. To achieve a desirable catalyst loading, a certain volume of the catalyst ink was carefully dropped to a well-polished glassy carbon (GC) rotating disk electrode (RDE). In order to prevent the contamination of catalysts, the GC electrode was polished using 0.05 μm alumina slurry, rinsed with deionized water, and then ultrasonicated in water for 30 s (3 times) and washed with distilled water until the appearance of a mirror-like surface on the GC electrode. Finally, the nonprecious catalyst loading was 0.510 mg·cm$^{-2}$, and the loading of Pt/C was 0.127 mg·cm$^{-2}$.

Electrochemical measurements were carried out in a three-electrode system on a Pine biopotentiostat electrochemical workstation (Pine Instrument Co., Durham, NC, USA) in 0.1 M KOH electrolyte. A rotating disk electrode (RDE) with a GC disk of 5 mm in diameter was used as the substrate for the working electrode, and a carbon rod was used as the counter electrode. The potential was recorded by a Ag/AgCl (saturated KCl solution) electrode as the reference electrode. Oxygen flow was used through the electrolyte in the cell for 30 min to saturate it with O2 before a measurement, and all electrochemical experiments were conducted in 0.1 M KOH at room temperature. The cyclic voltammetry (CV), linear sweep voltammetry (LSV), accelerated durability and chronoamperometric tests were carried out to measure the electrochemical performance of the catalyst. The

measured potentials vs. Ag/AgCl were converted to a reversible hydrogen electrode (RHE) scale according to the Nernst equation:

$$E_{RHE}(V) = E_{Ag/AgCl} + E^o_{Ag/AgCl} + 0.059pH$$

where $E^o_{Ag/AgCl}$ = 0.1976 V at 25 °C, $E_{Ag/AgCl}$ is the experimentally measured potential against the Ag/AgCl reference electrode, and ERHE is the converted potential vs. RHE.

CV experiments measure the electrochemical ability by recording the current variation of oxidation reduction reaction within a certain potential range. The electrolyte was saturated with oxygen before each experiment, the potential zone of CV tests were between 0 and 1.2 V (vs. RHE), the scan rate was 50 mV·s$^{-1}$, and at least 10 cycles were performed before collecting the data. During the experiment, a flow of $O_2$ was maintained over the electrolyte to ensure the $O_2$ saturation. In addition, tests were also performed in $N_2$ saturated electrolyte with a scan rate of 50 mV·s$^{-1}$ to measure the background current. Linear sweep voltammetry (LSV) tests were measured at various rotating speed from 400 to 2500 rpm with a scan rate of 10 mV·s$^{-1}$ in $O_2$-saturated electrolyte. An accelerated durability test (ADT) was carried out by cycling the catalysts between 0.6 and 1.05 V (vs. RHE) at a scan rate of 50 mV·s$^{-1}$. To evaluate the stability of the catalyst, the LSV curve of the catalyst was measured after 5000 cycles and compared with its initial LSV curve. A chronoamperometric test (CA) was performed to test the methanol tolerance and the electrochemical durability of the catalysts. At a constant potential (0.8 V vs. RHE), the decline rate of the current was recorded after 12,000 s. For the methanol tolerance, 3 mL of methanol was injected into the electrolyte at 200 s.

The electron transfer number (n) and kinetic current density (Jk) were examined by the Koutecky–Levich equation:

$$\frac{1}{j} = \frac{1}{j_l} + \frac{1}{j_k} = \frac{1}{B\omega^{\frac{1}{2}}} + \frac{1}{j_k}$$

$$B = 0.62nFC_0D_0^{\frac{2}{3}}V^{-\frac{1}{6}}$$

where $j$ is the measured current density, $j_k$ and $j_l$ are the kinetic and limiting current densities, $\omega$ is the angular velocity of the disk, $n$ is the electron transfer number, $F$ is the Faraday constant (96485 C·mol$^{-1}$), $C_0$ is the bulk concentration of $O_2$ (1.2 × 10$^{-6}$ mol·cm$^{-3}$), D$_0$ is the diffusion coefficient of $O_2$ (1.9 × 10$^{-5}$cm$^2$·s$^{-1}$), and $V$ is the kinematic viscosity of the electrolyte (0.01 cm$^2$·s$^{-1}$).

*3.4. Preparation of SA-Fe@NC*

Typically, 25.6 g of naphthalene and 11.7 g of indole were ground and placed in a 200 mL three-necked round-bottom flask, which was followed by the addition of powdered FeCl$_3$ (anhydrous). The mixture was stirred under nitrogen atmosphere at 85 °C for 4 h. Then, at room temperature, the solidified black product was transferred into a ceramic boat and placed in a tube furnace. Samples were heated to 800 °C, 950 °C and 1100 °C respectively with a heating rate of 5 °C min$^{-1}$ and kept at 800 °C, 950 °C and 1100 °C, respectively, for 3 h under the flowing argon gas (150 mL· min$^{-1}$) and then naturally cooled to room temperature. The obtained black powder were leached by 0.4 M HCl at 60 °C for 12 h, and the finally obtained black powder were named as SA-Fe@NC$_{800}$, SA-Fe@NC$_{950}$ and SA-Fe@NC$_{1100}$, respectively. The control sample was prepared at the same condition with AlCl$_3$ as catalyst and pyrolyzed at 800 °C. The sample SA-Fe@NC$_{800-1}$ was prepared from the precursor with 25.6 g of naphthalene and 4.68 g of indole in the presence of catalyst FeCl$_3$, which was followed by the pyrolysis at 800 °C and acid leaching. The sample SA-Fe@NC$_{800-2}$ was prepared from the precursor with only 11.7 g indole in the presence of catalyst FeCl$_3$, which was followed by the pyrolysis at 800 °C and acid leaching.

## 4. Conclusions

An efficient and versatile strategy was developed for the fabrication of atomically dispersed iron catalyst on carbonaceous materials. Lewis acid $FeCl_3$ plays multiple roles, including polymerization catalyst (the Lewis acid), pore-forming agent and iron source. With the assistance of comprehensive characterization techniques including XPS, HRTEM and XAFS, the atomically dispersed Fe active sites with an optimized coordination environment in the hierarchical porous carbon matrix are confirmed. The obtained SA-Fe@NC outperforms the commercial Pt/C catalyst in terms of ORR activity, tolerance of methanol crossover and long-term stability. This strategy could be employed in building Fe/N-modified carbon materials from fossil feedstock for potential large-scale commercial applications.

**Supplementary Materials:** The following supporting information can be downloaded at: https://www.mdpi.com/article/10.3390/catal12121608/s1, Figure S1: FT-ICR-MS results under the atmospheric pressure photo ionization (APPI) mode, (a) peaks show the presence of the naphthalene polymers and (b) peaks show the presence of the dehydrogenized co-polymers of indole–naphthalene; Figure S2: BET results for the control sample using $AlCl_3$ as polymerization catalyst; Figure S3: Raman of SA-Fe@NC$_{950}$; Figure S4: Raman of SA-Fe@NC$_{1100}$; Figure S5: XPS survey scan spectra of SA-Fe@NC$_{950}$, SA-Fe@NC$_{1100}$, SA-Fe@NC$_{800-1}$ and SA-Fe@NC$_{800-2}$; Figure S6: XPS of SA-Fe@NC$_{950}$; Figure S7: XPS of SA-Fe@NC$_{1100}$; Figure S8: SEM of SA-Fe@NC$_{800}$ showing structures with hierarchical pores; Figure S9: (a) CV curves of SA-Fe@NC$_{800}$, SA-Fe@NC$_{950}$, SA-Fe@NC$_{1100}$ and Pt/C catalysts in $O_2$-saturated (solid line) and $N_2$-saturated (dashed line) in 0.1 M KOH solutions at a sweep rate of 50 mV s$^{-1}$. (b) LSV curves of above catalysts in $O_2$-saturated 0.1 M KOH electrolyte at 10 mV s$^{-1}$ with a rotation rate of 1600 rpm; Figure S10: (a) CV curves of SA-Fe@NC$_{800-1}$, SA-Fe@NC$_{800-2}$ and Pt/C catalysts in $O_2$-saturated (solid line) and $N_2$-saturated (dashed line) in 0.1 M KOH solutions at a sweep rate of 50 mV s$^{-1}$. (b) LSV curves of above catalysts in $O_2$-saturated 0.1 M KOH electrolyte at 10 mV s$^{-1}$ with a rotation rate of 1600 rpm; Figure S11: LSV curves for Pt/C before and after 5000 CV cycles between 0.6 and 1.0 V; Figure S12: Stability test of the SA-Fe@NC$_{800}$ and the commercial Pt/C in the $O_2$-saturated 0.1 M KOH solution at 0.8 V under a rotation rate of 1600 rpm; Table S1: Elemental compositions of SA-Fe@NC catalysts determined by XPS.; Table S2: N contents of SA-Fe@NC catalysts pyrolyzed at different temperatures obtained from XPS N 1s spectra; Table S3: Structural parameters extracted from the Fe K-edge EXAFS fitting.

**Author Contributions:** Formal analysis, M.W., H.P., Q.P., X.G., W.W., H.W., J.S., X.Q.; writing—original draft preparation, Y.W.; writing—review and editing, Y.W.; Validation, W.Y.; supervision, D.M., Z.D., H.D.; project administration, Z.D., H.D. All authors have read and agreed to the published version of the manuscript.

**Funding:** This work received financial support from the Natural Science Foundation of China (22005007, 21725301, 21932002, 21821004), and the National Key R&D Program of China (2019YFB1505000, 2017YFB0602200).

**Data Availability Statement:** Not applicable.

**Conflicts of Interest:** The authors declare no conflict of interest.

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
