# Peer review of "Single-Atom Iron Catalyst Based on Functionalized Mesophase Pitch Exhibiting Efficient Oxygen Reduction Reaction Activity"

_catalysts, doi:10.3390/catal12121608_

Round 1
Reviewer 1 Report
Gu et al. prepared an atomically dispersed iron catalyst via the pyrolysis of a functionalized mesophase pitch (SA-Fe@NC) that displayed excellent oxygen reduction reaction activity (E1/2 = 0.89 V) that outperforms the commercial Pt/C catalyst (E1/2 = 0.85 V). All the discussions and data analysis in the manuscript seems interesting, and supports the authors’ conclusion. Thus, I strongly recommend to publish this paper. However, a minor revision is needed.
1. Authors must define the term before using abbreviation (see in abstract, SA).
2. What is the role of Fe during the naphthalene and indole oligomerization, catalyst? And what was the ratio of all precursors?
3. Authors can prepare NC material, as counter sample) using similar precursors (without Fe) as well as experimental condition? Then compare the structural data of NC and SA-Fe@NC, to find the role of Fe.
4. Why authors applied 800 °C pyrolysis temperature? samples were also prepared other temperatures?
5. What was the loading of catalysts, used for the electrocatalytic ORR?
6. From LSV curve of Pt/C catalyst, it is hard to assume E1/2=0.85 V, please check it again.
7. What was the ORR active site in the SA-Fe@NC sample, can authors add the SCN poising test?
Author Response
Response to the comments made by the reviewer
Comments
Gu et al. prepared an atomically dispersed iron catalyst via the pyrolysis of a functionalized mesophase pitch (SA-Fe@NC) that displayed excellent oxygen reduction reaction activity (E1/2 = 0.89 V) that outperforms the commercial Pt/C catalyst (E1/2 = 0.85 V). All the discussions and data analysis in the manuscript seems interesting, and supports the authors’ conclusion. Thus, I strongly recommend to publish this paper. However, a minor revision is needed.
Response:
Thank you for your valuable comments and suggestions. Following is the point-by-point response for reviewers' suggestions. All changes in the revised version are highlighted(see attachment for the updated manuscript), and the relevant page and line numbers are mentioned.
- Authors must define the term before using abbreviation (see in abstract, SA).
Response: updated.
- What is the role of Fe during the naphthalene and indole oligomerization, catalyst? And what was the ratio of all precursors?
Response: Fe in the form of FeCl3 is a Lewis acid, oligomerize the monomers. It was added in the molar ratio between 0.1 to 0.4 with respect to the monomers.
- Authors can prepare NC material, as counter sample) using similar precursors (without Fe) as well as experimental condition? Then compare the structural data of NC and SA-Fe@NC, to find the role of Fe.
Response: Without Fe(FeCl3), no oligomerization would happen, thus no NC materials would form after pyrolysis. We did use AlCl3 (another Lewis acid) to prepare the precursor for the carbon materials, but it was just carbon afterwards, with no electrocatalytic functions.
- Why authors applied 800 °C pyrolysis temperature? samples were also prepared other temperatures?
Response: 800 °C is an optimized condition for this materials, higher temperatures were conducted (for example above 1100°C, see SI), however, they showed less electrocatalytic performance in the test.
- What was the loading of catalysts, used for the electrocatalytic ORR?
Response: the loading of catalyst is 0.510 mg·cm−2 (Described in the SI)
- From LSV curve of Pt/C catalyst, it is hard to assume E1/2=0.85 V, please check it again.
Response: E1/2=0.84 V, updated.
- What was the ORR active site in the SA-Fe@NC sample, can authors add the SCN poising test?
Response: The ORR active site is the Fe-Nx complex dispersed with the carbon matrix, where x is believed to be 3-4. The SCN molecule would bind to the metal spontaneously and decrease the ORR activity, this could be found in the literature. (J. Am. Chem. Soc. 2021, 143, 7819−7827 )

Reviewer 2 Report
This manuscript describes a highly active Fe-N-C non-Pt catalyst for the ORR in alkaline media synthesized by a simple pyrolysis of pre-mixed naphthalene, indole and FeCl3. The authors claim that (1) a versatile strategy was developed for the fabrication of atomically dispersed iron catalyst on carbonaceous materials (2) the atomically dispersed Fe forms active site of Fe-N4 coordination structure (3) the obtained SA-Fe@NC outperforms the commercial Pt/C catalyst in terms of ORR activity, tolerance of methanol crossover and long-term stability.
However, extensive studies have been conducted on the Fe-N-C non-Pt catalysts and it was demonstrated that (1) the Fe single atom exists in the Fe-N-C non-Pt catalysts [e.g., H. T. Chung et al., Science, 357, 479-484 (2017). Y. Nabae et al., Catal. Sci. Technol., 10, 493-501 (2020).], (2) the ORR activity of the Fe-N-C non-Pt catalyst increases with the increase in the pyrolysis temperature and declines beyond a certain temperature, which is due to decomposition of the active site and multi-step pyrolysis is effective to enhance the ORR activity [e.g., Y. Nabae et al., Catal. Sci. Technol., 4, 1400-1406 (2014).], (3) these non-Pt catalysts are tolerant to CO and methanol [e.g., K. Gong et al., Science, 323, 760-264 (2009).].
Therefore, although the precursors use in this manuscript are model for the fossil feedstock, this reviewer feels that there is little new specific finding in this manuscript and it is difficult to publish this study in Catalysts.
Author Response
Thank you for your valuable comments.
We acknowledge that there are tremendous work on non-PGM ORR catalysts in recent years, yet from the perspective of application and manufacturing, as a synthetic group, we aim to develop more effective ways of fabricating these materials, with relatively less cost. Meanwhile, although mesophase is an interesting precursor for carbon material, the functionalization of mesophase by introducing hetro-atoms were not extensively exploited, as traditionally, mesophase was strictly required as high purity and high crystallinity. Our means brings the new opportunity for value-added carbonaceous materials, in the mean time, taking advantage of the versatility of petroleum feedstock. From academic perspective, in our recent work, we have synthesized S doped FeN@C materials, the pathway of sulphur doping through oilgomerization has significant effects on the Fe-N active sites as well as the overall ORR activity and stability. Therefore, it is necessary to further investigate the pathway of precursor formation and its influence to the resulted materials.
Reviewer 3 Report
In this manuscript, Gu et al. reported the facile synthesis of an atomically dispersed iron catalyst via the pyrolysis of a functionalized mesophase pitch (obtained from the petroleum by-products), and its application as a high-efficiency electrocatalyst for oxygen reduction reaction (ORR) in alkaline medium. The as-developed catalyst exhibited superior activity, stability and methanol tolerance to the benchmark Pt/C catalyst, showing promise for practical applications in fuel cells. This work provides a novel means towards active and low-cost ORR catalysts using abundant petroleum resources. Overall, the idea has good novelty and the manuscript was well drafted in a concise manner. Based on the expertise of this reviewer, I would like to support its publication at Catalysts. To further improve the manuscript, the below minor points need to be addressed.
1. While the manuscript focuses on single atom catalysts, the Introduction section did not provide sufficient background information on single atom catalysts. It is suggested that the authors provide some background information about single atom catalysts. Related works are suggested to be referenced, such as Materials Reports: Energy, 2022, 2, 100144.
2. How potential vs. Ag/AgCl was converted to that vs. RHE should be detailed in Experimental section.
3. As seen from Figure 4 and relevant discussion, FeO was another reference sample of Fe XANES data, but was not mentioned in the Experimental section in Supporting Information.
4. To appeal to a broader readership, very recent works on ORR catalysts (e.g., Small, 2022, 18, 2105329; Energy Fuels 2021, 35, 13585-13609) are suggested to be referenced in Introduction section.
5. Did the authors analyse the high-resolution XPS data for Fe 2p? Would it give some useful information about the bonding of Fe-Nx?
6. Some technical terms, such as E1/2, RHE, etc., when they first appear, should be defined.
7. Figure 2a and S2, y axis, “Absorbed” should be revised into “Adsorbed”.
Author Response
Response to the comments made by the reviewer
Comments
In this manuscript, Gu et al. reported the facile synthesis of an atomically dispersed iron catalyst via the pyrolysis of a functionalized mesophase pitch (obtained from the petroleum by-products), and its application as a high-efficiency electrocatalyst for oxygen reduction reaction (ORR) in alkaline medium. The as-developed catalyst exhibited superior activity, stability and methanol tolerance to the benchmark Pt/C catalyst, showing promise for practical applications in fuel cells. This work provides a novel means towards active and low-cost ORR catalysts using abundant petroleum resources. Overall, the idea has good novelty and the manuscript was well drafted in a concise manner. Based on the expertise of this reviewer, I would like to support its publication at Catalysts. To further improve the manuscript, the below minor points need to be addressed.
Response:
Thank you for your valuable comments and suggestions. Following is the point-by-point response for reviewers' suggestions. All changes in the revised version are highlighted, and the relevant page and line numbers are mentioned.
- While the manuscript focuses on single atom catalysts, the Introduction section did not provide sufficient background information on single atom catalysts. It is suggested that the authors provide some background information about single atom catalysts. Related works are suggested to be referenced, such as Materials Reports: Energy, 2022, 2, 100144.
Response: Related reference were updated in the maintext.
- How potential vs. Ag/AgCl was converted to that vs. RHE should be detailed in Experimental section.
Response: “The measured potentials vs. Ag/AgCl were converted to a reversible hydrogen electrode (RHE) scale according to the Nernst equation:
ERHE (V) = EAg/AgCl + Eo Ag/AgCl + 0.059 × pH (1)
where Eo Ag/AgCl = 0.1976 V at 25 °C, EAg/AgCl is the experimentally measured potential against the Ag/AgCl reference electrode, and ERHE is the converted potential vs. RHE.” is updated in the Methods
- As seen from Figure 4 and relevant discussion, FeO was another reference sample of Fe XANES data, but was not mentioned in the Experimental section in Supporting Information.
Response: updated in the SI
- To appeal to a broader readership, very recent works on ORR catalysts (e.g., Small, 2022, 18, 2105329; Energy Fuels 2021, 35, 13585-13609) are suggested to be referenced in Introduction section.
Response: updated
- Did the authors analyse the high-resolution XPS data for Fe 2p? Would it give some useful information about the bonding of Fe-Nx?
Response: High-resolution XPS characterization for Fe 2p was attempted, however, since the loading of Fe in the carbon matrix is too low, the XPS background noise is significant, it might be controversial to argue the coordination number of Fe. So we choose to use XANES data instead.
- Some technical terms, such as E1/2, RHE, etc., when they first appear, should be defined.
Response: updated
- Figure 2a and S2, y axis, “Absorbed” should be revised into “Adsorbed”.
Response: updated
Round 2
Reviewer 2 Report
As this reviewer commented in the first reviewing, there are little new and attractive insights and findings on the Fe-N-C non-Pt catalyst for the ORR in this manuscript. The authors just changed precursors which may be abundant in the oil companies (the fossil feedstock) and cost effective. However, as this reviewer showed the references in the first reviewing, all the researchers on this field do know the contents (see below). Therefore, this reviewer does not recommend publication of this manuscript to this journal of Catalysts.
K. Gong et al., Science, 323, 760-264 (2009).
Y. Nabae et al., Catal. Sci. Technol., 4, 1400-1406 (2014).
H. T. Chung et al., Science, 357, 479-484 (2017).
Y. Nabae et al., Catal. Sci. Technol., 10, 493-501 (2020).
Author Response
We thanks for reviewer’s kind comments and we think our work is worthy for its publication in the journal of Catalysts due to the following reasons. Regarding the novelties queries, we disagree about the claim “just change a precusor...”, since what we focused on is the preparation of functional precusors (mesophase) through facile oligomerization, which was characterized revealing interesting structures in ICR-MS. To our best knowledge, the pathway is still one of the few in all publisehed ORR Fe/N/C catalysts. One of the lastest paper in referred in 2020, the ORR Fe/N/C was prepared by the polyimide and Fe salt. Here valuable monomers were used and Fe salt was blended arbitarily, yet the resulted high-cost precusor was pyrolyzed and gave similar Fe/N/C active sites. In our approach, N-containing molecules (abundant in oil) could be easily oligomerized and dispersed into carbon matrix, and other hetroatom-containing molecules that undergo Friedel-Crafts reaction also applies. Our means could bring more functionalities to the carbonaceous precusors, in the mean time, taking advantage of the versatility of petroleum feedstock. Therefore from the synthetic perspective, it is necessary to further investigate the pathways of functional mesophase formation and our work is an important attempt.